# Perceptual Generative Autoencoders

## Abstract

Modern generative models are usually designed to match target distributions directly in the data space, where the intrinsic dimensionality of data can be much lower than the ambient dimensionality. We argue that this discrepancy may contribute to the difficulties in training generative models. We therefore propose to map both the generated and target distributions to the latent space using the encoder of a standard autoencoder, and train the generator (or decoder) to match the target distribution in the latent space. The resulting method, perceptual generative autoencoder (PGA), is then incorporated with a maximum likelihood or variational autoencoder (VAE) objective to train the generative model. With maximum likelihood, PGAs generalize the idea of reversible generative models to unrestricted neural network architectures and arbitrary latent dimensionalities. When combined with VAEs, PGAs can generate sharper samples than vanilla VAEs. Compared to other autoencoder-based generative models using simple priors, PGAs achieve state-of-the-art FID scores on CIFAR-10 and CelebA.

## 1 Introduction

Recent years have witnessed great interest in generative models, mainly due to the success of generative adversarial networks (GANs) (Goodfellow et al., 2014; Radford et al., 2016; Karras et al., 2018; Brock et al., 2019). Despite their prevalence, the adversarial nature of GANs can lead to a number of challenges, such as unstable training dynamics and mode collapse. Since the advent of GANs, substantial efforts have been devoted to addressing these challenges (Salimans et al., 2016; Arjovsky et al., 2017; Gulrajani et al., 2017; Miyato et al., 2018), while non-adversarial approaches that are free of these issues have also gained attention. Examples include variational autoencoders (VAEs) (Kingma & Welling, 2014), reversible generative models (Dinh et al., 2014; 2017; Kingma & Dhariwal, 2018), and Wasserstein autoencoders (WAEs) (Tolstikhin et al., 2018).

However, non-adversarial approaches often have significant limitations. For instance, VAEs tend to generate blurry samples, while reversible generative models require restricted neural network architectures or solving neural differential equations (Grathwohl et al., 2019). Furthermore, to use the change of variable formula, the latent space of a reversible model must have the same dimensionality as the data space, which is unreasonable considering that real-world, high-dimensional data (e.g., images) tends to lie on low-dimensional manifolds, and thus results in redundant latent dimensions and variability. Intriguingly, recent research (Arjovsky et al., 2017; Dai & Wipf, 2019) suggests that the discrepancy between the intrinsic and ambient dimensionalities of data also contributes to the difficulties in training GANs and VAEs.

In this work, we present a novel framework for training autoencoder-based generative models, with non-adversarial losses and unrestricted neural network architectures. Given a standard autoencoder and a target data distribution, instead of matching the target distribution in the data space, we map both the generated and target distributions to the latent space using the encoder, and train the generator (or decoder) to minimize the divergence between the mapped distributions. We prove, under mild assumptions, that by minimizing a form of latent reconstruction error, matching the target distribution in the latent space implies matching it in the data space. We call this framework *perceptual generative autoencoder (PGA)*. We show that PGAs enable training generative autoencoders with maximum likelihood, without restrictions on architectures or latent dimensionalities. In addition, when combined with VAEs, PGAs can generate sharper samples than vanilla VAEs.[1]

---

[1] Code is available at `https://bit.ly/2U0kRYL`.

We summarize our main contributions as follows:

- A training framework, PGA, for generative autoencoders is developed to match the target distribution in the latent space, which, we prove, ensures the matching in the data space.
- We combine the PGA framework with maximum likelihood, and remove the restrictions of reversible generative models on neural network architectures and latent dimensionalities.
- We combine the PGA framework with VAE, which solves the problem of blurry samples, without introducing any auxiliary models or sophisticated model architectures.

## 2 RELATED WORK

Autoencoder-based generative models are trained by minimizing an data reconstruction loss with regularizations. As an early approach, denoising autoencoders (DAEs) (Vincent et al., 2008) are trained to recover the original input from an intentionally corrupted input. Then a generative model can be obtained by sampling from a Markov chain (Bengio et al., 2013). To sample from a decoder directly, most recent approaches resort to mapping a simple prior distribution to a data distribution using the decoder. For instance, adversarial autoencoders (AAEs) (Makhzani et al., 2016) and Wasserstein autoencoders (WAEs) (Tolstikhin et al., 2018) attempt to match the aggregated posterior and the prior, either by adversarial training or by minimizing their Wasserstein distance. However, due to the use of deterministic encoders, there can be "holes" in the latent space that are not covered by the aggregated posterior, which would result in poor sample quality (Rubenstein et al., 2018). By using stochastic encoders and variational inference, variational autoencoders (VAEs) are likely to suffer less from this problem, but are known to generate blurry samples (Rezende & Viola, 2018; Dai & Wipf, 2019). Nevertheless, as we will show, the latter problem can be addressed by moving the VAE reconstruction loss from the data space to the latent space.

In a different line of work, reversible generative models (Dinh et al., 2014; 2017; Kingma & Dhariwal, 2018) are developed to enable exact inference. Consequently, by the change of variables theorem, the likelihood of each data sample can be exactly computed and optimized. However, to avoid expensive Jacobian determinant computations, reversible models can only be composed of restricted transformations, rather than general neural network architectures. While this restriction can be relaxed by utilizing recently developed neural ordinary differential equations (Chen et al., 2018; Grathwohl et al., 2019), they still rely on a shared dimensionality between latent and data spaces, which remains an unnatural restriction. In this work, we use the proposed training framework to trade exact inference for unrestricted neural network architectures and arbitrary latent dimensionalities, generalizing maximum likelihood training to autoencoder-based models.

## 3 METHODS

### 3.1 PERCEPTUAL GENERATIVE MODEL

Let $f_\phi : \mathbb{R}^D \to \mathbb{R}^H$ be the encoder parameterized by $\phi$, and $g_\theta : \mathbb{R}^H \to \mathbb{R}^D$ be the decoder parameterized by $\theta$. Our goal is to obtain a decoder-based generative model, which maps a simple prior distribution to a target data distribution, $\mathcal{D}$. A summary of notations is provided in Appendix A. Throughout this paper, we use $\mathcal{N}(\mathbf{0}, \mathbf{I})$ as the prior distribution.

For $\mathbf{z} \in \mathbb{R}^H$, the output of the decoder, $g_\theta(\mathbf{z})$, lies in a manifold that is at most $H$-dimensional. Therefore, if we train the autoencoder to minimize

$$L_r = \frac{1}{2} \mathbb{E}_{\mathbf{x} \sim \mathcal{D}} \left[ \|\hat{\mathbf{x}} - \mathbf{x}\|_2^2 \right], \tag{1}$$

where $\hat{\mathbf{x}} = g_\theta(f_\phi(\mathbf{x}))$, then $\hat{\mathbf{x}}$ can be seen as a projection of the input data, $\mathbf{x}$, onto the manifold of $g_\theta(\mathbf{z})$. Let $\hat{\mathcal{D}}$ denote the reconstructed data distribution, i.e., $\hat{\mathbf{x}} \sim \hat{\mathcal{D}}$. Given enough capacity of the encoder, $\hat{\mathcal{D}}$ is the best approximation to $\mathcal{D}$ (in terms of $\ell_2$-distance), that we can obtain from the decoder, and thus can serve as a surrogate target distribution for training the decoder-based generative model.

Due to the difficulty in directly matching the generated distribution with the data-space target distribution, $\hat{\mathcal{D}}$, we reuse the encoder to map $\hat{\mathcal{D}}$ to a latent-space target distribution, $\hat{\mathcal{H}}$. We then

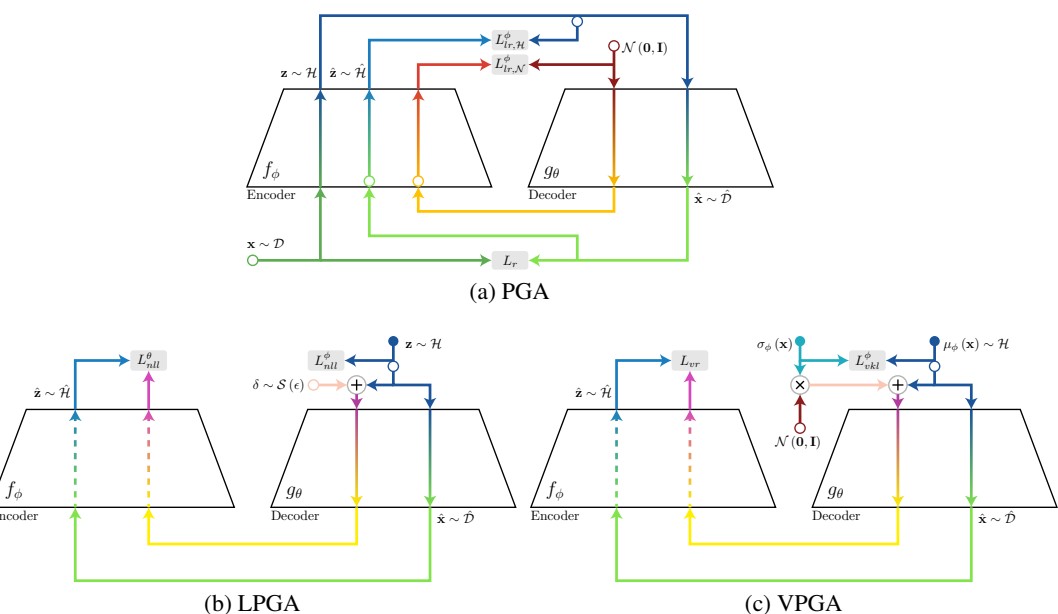

Figure 1: Illustration of the training process of PGAs. The overall loss function consists of (a) the basic PGA losses, and either (b) the LPGA-specific losses or (c) the VPGA-specific losses. Circles indicate where the gradient is truncated, and dashed lines indicate where the gradient is ignored when updating parameters.

transform the problem of matching $\hat{\mathcal{D}}$ in the data space into matching $\hat{\mathcal{H}}$ in the latent space. In other words, we aim to ensure that for $\mathbf{z} \sim \mathcal{N}(\mathbf{0}, \mathbf{I})$, if $f_\phi(g_\theta(\mathbf{z})) \sim \hat{\mathcal{H}}$, then $g_\theta(\mathbf{z}) \sim \hat{\mathcal{D}}$. In the following, we define $h = f_\phi \circ g_\theta$ for notational convenience.

To this end, we minimize the following latent reconstruction loss w.r.t. $\phi$:

$$L_{lr,\mathcal{N}}^\phi = \frac{1}{2}\mathbb{E}_{\mathbf{z}\sim\mathcal{N}(\mathbf{0},\mathbf{I})}\left[\|h(\mathbf{z}) - \mathbf{z}\|_2^2\right]. \tag{2}$$

Let $Z(\mathbf{x})$ be the set of all $\mathbf{z}$'s that are mapped to the same $\mathbf{x}$ by $g_\theta$, we have the following theorem:

**Theorem 1.** *Assuming $\mathbb{E}[\mathbf{z} \mid \mathbf{x}] \in Z(\mathbf{x})$ for all $\mathbf{x}$ generated by $g_\theta$, and sufficient capacity of $f_\phi$; for $\mathbf{z} \sim \mathcal{N}(\mathbf{0}, \mathbf{I})$, if Eq. (2) is minimized and $h(\mathbf{z}) \sim \hat{\mathcal{H}}$, then $g_\theta(\mathbf{z}) \sim \hat{\mathcal{D}}$.*

We defer the proof to Appendix B.1. Note that Theorem 1 requires that different $\mathbf{x}$'s generated by $g_\theta$ (from $\mathcal{N}(\mathbf{0}, \mathbf{I})$ and $\mathcal{H}$) are mapped to different $\mathbf{z}$'s by $f_\phi$. In theory, minimizing Eq. (2) would suffice, since $\mathcal{N}(\mathbf{0}, \mathbf{I})$ is supported on the whole $\mathbb{R}^H$. However, there can be $\mathbf{z}$'s with low probabilities in $\mathcal{N}(\mathbf{0}, \mathbf{I})$, but with high probabilities in $\mathcal{H}$ that are not well covered by Eq. (2). Therefore, it is sometimes helpful to minimize another latent reconstruction loss on $\mathcal{H}$:

$$L_{lr,\mathcal{H}}^\phi = \frac{1}{2}\mathbb{E}_{\mathbf{z}\sim\mathcal{H}}\left[\|h(\mathbf{z}) - \mathbf{z}\|_2^2\right]. \tag{3}$$

In practice, we observe that $L_{lr,\mathcal{H}}^\phi$ is often small without explicit minimization, which we attribute to its consistency with the minimization of $L_r$.

By Theorem 1, the problem of training the generative model reduces to training $h$ to map $\mathcal{N}(\mathbf{0}, \mathbf{I})$ to $\hat{\mathcal{H}}$, which we refer to as the perceptual generative model. In the subsequent subsections, we present a maximum likelihood approach, a VAE-based approach, and a unified approach to train the perceptual generative model. The basic loss function of PGAs is given by

$$L_{pga} = L_r + \alpha L_{lr,\mathcal{N}}^\phi + \beta L_{lr,\mathcal{H}}^\phi, \tag{4}$$

where $\alpha$ and $\beta$ are hyperparameters to be tuned. Eq. (4) is also illustrated in Fig. 1a.

### 3.2 A Maximum Likelihood Approach

We first assume the invertibility of $h$. For $\hat{\mathbf{x}} \sim \hat{\mathcal{D}}$, let $\hat{\mathbf{z}} = f_\phi(\hat{\mathbf{x}}) \sim \hat{\mathcal{H}}$. We can train $h$ directly with maximum likelihood using the change of variables formula as

$$\mathbb{E}_{\hat{\mathbf{z}} \sim \hat{\mathcal{H}}}[\log p(\hat{\mathbf{z}})] = \mathbb{E}_{\mathbf{z} \sim \mathcal{H}}\left[\log p(\mathbf{z}) - \log\left|\det\left(\frac{\partial h(\mathbf{z})}{\partial \mathbf{z}}\right)\right|\right]. \tag{5}$$

Ideally, we would like to maximize Eq. (5) only w.r.t. the parameters of the generative model (i.e., $\theta$). However, directly optimizing the first term in Eq. (5) requires computing $\mathbf{z} = h^{-1}(\hat{\mathbf{z}})$, which is usually unknown. Nevertheless, for $\hat{\mathbf{z}} \sim \hat{\mathcal{H}}$, we have $h^{-1}(\hat{\mathbf{z}}) = f_\phi(\mathbf{x})$ and $\mathbf{x} \sim \mathcal{D}$, and thus we can minimize the following loss function w.r.t. $\phi$ instead:

$$L_{nll}^\phi = -\mathbb{E}_{\mathbf{z} \sim \mathcal{H}}[\log p(\mathbf{z})] = \frac{1}{2}\mathbb{E}_{\mathbf{x} \sim \mathcal{D}}\left[\|f_\phi(\mathbf{x})\|_2^2\right]. \tag{6}$$

To avoid computing the Jacobian in the second term of Eq. (5), which is slow for unrestricted architectures, we approximate the Jacobian determinant and derive a loss function to be minimized w.r.t. $\theta$:

$$L_{nll}^\theta = \frac{H}{2}\mathbb{E}_{\mathbf{z} \sim \mathcal{H}, \delta \sim \mathcal{S}(\epsilon)}\left[\log \frac{\|h(\mathbf{z} + \delta) - h(\mathbf{z})\|_2^2}{\|\delta\|_2^2}\right] \approx \mathbb{E}_{\mathbf{z} \sim \mathcal{H}}\left[\log\left|\det\left(\frac{\partial h(\mathbf{z})}{\partial \mathbf{z}}\right)\right|\right], \tag{7}$$

where $\mathcal{S}(\epsilon)$ can be either $\mathcal{N}(\mathbf{0}, \epsilon^2\mathbf{I})$, or a uniform distribution on a small $(H-1)$-sphere of radius $\epsilon$ centered at the origin. The latter choice is expected to introduce slightly less variance. We show below that the approximation gives an upper bound when $\epsilon \to 0$. Eqs. (6) and (7) are illustrated in Fig. 1b.

**Proposition 1.** *For $\epsilon \to 0$,*

$$\log\left|\det\left(\frac{\partial h(\mathbf{z})}{\partial \mathbf{z}}\right)\right| \leq \frac{H}{2}\mathbb{E}_{\delta \sim \mathcal{S}(\epsilon)}\left[\log \frac{\|h(\mathbf{z} + \delta) - h(\mathbf{z})\|_2^2}{\|\delta\|_2^2}\right]. \tag{8}$$

*The inequality is tight if $h$ is a multiple of the identity function around $\mathbf{z}$.*

We defer the proof to Appendix B.2. We note that the above discussion relies on the invertibility of $h$, which, however, is not required by the resulting method. Indeed, when $h$ is invertible at some point $\mathbf{z}$, the latent reconstruction loss ensures that $h$ is close to the identity function around $\mathbf{z}$, and hence the tightness of the upper bound in Eq. (8). Otherwise, when $h$ is not invertible at some $\mathbf{z}$, the logarithm of the Jacobian determinant at $\mathbf{z}$ becomes infinite, in which case Eq. (5) cannot be optimized. Nevertheless, since $\|h(\mathbf{z} + \delta) - h(\mathbf{z})\|_2^2$ is unlikely to be zero if the model is properly initialized, the approximation in Eq. (7) remains finite, and thus can be optimized regardless.

To summarize, we train the autoencoder to obtain a generative model by minimizing the following loss function:

$$L_{lpga} = L_{pga} + \gamma\left(L_{nll}^\phi + L_{nll}^\theta\right). \tag{9}$$

We refer to this approach as maximum likelihood PGA (LPGA).

### 3.3 A VAE-based Approach

The original VAE is trained by maximizing the evidence lower bound on $\log p(\mathbf{x})$ as

$$\begin{aligned}
\log p(\mathbf{x}) &\geq \log p(\mathbf{x}) - \mathbb{KL}(q(\mathbf{z}' \mid \mathbf{x}) \,||\, p(\mathbf{z}' \mid \mathbf{x})) \\
&= \mathbb{E}_{\mathbf{z}' \sim q(\mathbf{z}' \mid \mathbf{x})}[\log p(\mathbf{x} \mid \mathbf{z}')] - \mathbb{KL}(q(\mathbf{z}' \mid \mathbf{x}) \,||\, p(\mathbf{z}')),
\end{aligned} \tag{10}$$

where $p(\mathbf{x} \mid \mathbf{z}')$ is modeled with the decoder, and $q(\mathbf{z}' \mid \mathbf{x})$ is modeled with the encoder. Note that $\mathbf{z}'$ denotes the stochastic version of $\mathbf{z}$, whereas $\mathbf{z}$ remains deterministic for the basic PGA losses in Eqs. (2) and (3). In our case, we would like to modify Eq. (10) in a way that helps maximize $\log p(\hat{\mathbf{z}})$. Therefore, we replace $p(\mathbf{x} \mid \mathbf{z}')$ on the r.h.s. of Eq. (10) with $p(\hat{\mathbf{z}} \mid \mathbf{z}')$, and derive a lower bound on $\log p(\hat{\mathbf{z}})$ as

$$\begin{aligned}
\log p(\hat{\mathbf{z}}) &\geq \log p(\hat{\mathbf{z}}) - \mathbb{KL}(q(\mathbf{z}' \mid \mathbf{x}) \,||\, p(\mathbf{z}' \mid \hat{\mathbf{z}})) \\
&= \mathbb{E}_{\mathbf{z}' \sim q(\mathbf{z}' \mid \mathbf{x})}[\log p(\hat{\mathbf{z}} \mid \mathbf{z}')] - \mathbb{KL}(q(\mathbf{z}' \mid \mathbf{x}) \,||\, p(\mathbf{z}')).
\end{aligned} \tag{11}$$

Similar to the original VAE, we make the assumption that $q\left(\mathbf{z}' \mid \mathbf{x}\right)$ and $p\left(\hat{\mathbf{z}} \mid \mathbf{z}'\right)$ are Gaussian; i.e., $q\left(\mathbf{z}' \mid \mathbf{x}\right) = \mathcal{N}\left(\mathbf{z}' \mid \mu_\phi\left(\mathbf{x}\right), \operatorname{diag}\left(\sigma_\phi^2\left(\mathbf{x}\right)\right)\right)$, and $p\left(\hat{\mathbf{z}} \mid \mathbf{z}'\right) = \mathcal{N}\left(\hat{\mathbf{z}} \mid \mu_{\theta,\phi}\left(\mathbf{z}'\right), \sigma^2\mathbf{I}\right)$. Here, $\mu_\phi\left(\cdot\right) = f_\phi\left(\cdot\right)$, $\mu_{\theta,\phi}\left(\cdot\right) = h\left(\cdot\right)$, and $\sigma > 0$ is a tunable scalar. Note that if $\sigma$ is fixed, the first term on the r.h.s. of Eq. (11) has a trivial maximum, where $\mathbf{z}$, $\hat{\mathbf{z}}$, and $\mu_{\theta,\phi}\left(\mathbf{z}'\right)$ are all close to zero. To circumvent this, we set $\sigma$ proportional to the $\ell_2$-norm of $\mathbf{z}$.

The VAE variant is trained by minimizing

$$L_{vae} = L_{vr} + L_{vkl}^\phi = -\mathbb{E}_{\mathbf{x}\sim\mathcal{D}}\left[\mathbb{E}_{\mathbf{z}'\sim q(\mathbf{z}'\mid\mathbf{x})}\left[\log p\left(\hat{\mathbf{z}}\mid\mathbf{z}'\right)\right] - \mathbb{KL}(q\left(\mathbf{z}'\mid\mathbf{x}\right)\mid\mid p\left(\mathbf{z}'\right))\right], \quad (12)$$

where $L_{vr}$ and $L_{vkl}^\phi$ are, respectively, the reconstruction and KL divergence losses of VAE, as illustrated in Fig. 1c. Accordingly, the overall loss function is given by

$$L_{vpga} = L_{pga} + \eta L_{vae}. \quad (13)$$

We refer to this approach as variational PGA (VPGA).

### 3.4 A High-level View of the PGA Framework

We summarize what each loss term achieves, and explain from a high-level how they work together.

**Data reconstruction loss** (Eq. (1)): For Theorem 1 to hold, we need to use the reconstructed data distribution ($\hat{\mathcal{D}}$), instead of the original data distribution ($\mathcal{D}$), as the target distribution. Therefore, minimizing the data reconstruction loss ensures that the target distribution is close to the data distribution.

**Latent reconstruction loss** (Eqs. (2) and (3)): The encoder ($f_\phi$) is reused to map data-space distributions to the latent space. As shown by Theorem 1, minimizing the latent reconstruction loss (w.r.t. the parameters of the encoder) ensures that if the generated distribution and the target distribution can be mapped to the same distribution ($\hat{\mathcal{H}}$) in the latent space by the encoder, then the generated distribution and the target distribution are the same.

**Maximum likelihood loss** (Eqs. (6) and (7)) or **VAE loss** (Eq. (12)): The decoder ($g_\theta$) and encoder ($f_\phi$) together can be considered as a perceptual generative model ($f_\phi \circ g_\theta$), which is trained to map $\mathcal{N}\left(\mathbf{0}, \mathbf{I}\right)$ to the latent-space target distribution ($\hat{\mathcal{H}}$) by minimizing either the maximum likelihood loss or the VAE loss.

The first loss allows to use the reconstructed data distribution as the target distribution. The second loss transforms the problem of matching the target distribution in the data space into matching it in the latent space. The latter problem is then solved by the third loss. Therefore, the three losses together ensure that the generated distribution is close to the data distribution.

### 3.5 A Unified Approach

While the loss functions of maximum likelihood and VAE seem completely different in their original forms, they share remarkable similarities when considered in the PGA framework (see Figs. 1b and 1c). Intuitively, observe that

$$L_{vkl}^\phi = L_{nll}^\phi + \frac{1}{2}\mathbb{E}_{\mathbf{x}\sim\mathcal{D}}\sum_{i\in[H]}\left[\sigma_{\phi,i}^2\left(\mathbf{x}\right) - \log\left(\sigma_{\phi,i}^2\left(\mathbf{x}\right)\right)\right], \quad (14)$$

which means both $L_{nll}^\phi$ and $L_{vkl}^\phi$ tend to attract the latent representations of data samples to the origin. In addition, $L_{nll}^\theta$ expands the volume occupied by each sample in the latent space, which can be also achieved by $L_{vr}$ with the second term of Eq. (14).

More concretely, we observe that both $L_{nll}^\theta$ and $L_{vr}$ are minimizing the difference between $h\left(\mathbf{z}\right)$ and $h\left(\mathbf{z}+\delta'\right)$, where $\delta'$ is some additive zero-mean noise. However, they differ in that the variance of $\delta'$ is fixed for $L_{nll}^\theta$, but is trainable for $L_{vr}$; and the distance between $h\left(\mathbf{z}\right)$ and $h\left(\mathbf{z}+\delta'\right)$ are defined in two different ways. In fact, $L_{vr}$ is a squared $\ell_2$-distance derived from the Gaussian assumption on $\hat{\mathbf{z}}$, whereas $L_{nll}^\theta$ can be derived similarly by assuming that $d^H = \|\hat{\mathbf{z}} - h\left(\mathbf{z}+\delta\right)\|_2^H$ follows a reciprocal distribution as

$$p\left(d^H; a, b\right) = \frac{1}{d^H\left(\log\left(b\right) - \log\left(a\right)\right)}, \quad (15)$$

where $a \leq d^H \leq b$, and $a > 0$. The exact values of $a$ and $b$ are irrelevant, as they only appear in an additive constant when we take the logarithm of $p\left(d^H; a, b\right)$.

Since there is no obvious reason for assuming Gaussian $\hat{\mathbf{z}}$, we can instead assume $\hat{\mathbf{z}}$ to follow the distribution defined in Eq. (15), and multiply $H$ by a tunable scalar, $\gamma'$, similar to $\sigma$. Furthermore, we can replace $\delta$ in Eq. (7) with $\delta' \sim \mathcal{N}\left(\mathbf{0}, \mathrm{diag}\left(\sigma_\phi^2\left(\mathbf{x}\right)\right)\right)$, as it is defined for VPGA with a subtle difference that here $\sigma_\phi^2\left(\mathbf{x}\right)$ is constrained to be greater than $\epsilon^2$. As a result, LPGA and VPGA are unified into a single approach, which has a combined loss function as

$$L_{lvpga} = L_{pga} + \gamma' L_{vr} + \gamma L_{nll}^\phi + \eta L_{vkl}^\phi. \tag{16}$$

When $\gamma' = \gamma$ and $\eta = 0$, Eq. (16) is equivalent to Eq. (9), considering that $\sigma_\phi^2\left(\mathbf{x}\right)$ will be optimized to approach $\epsilon^2$. Similarly, when $\gamma = 0$, Eq. (16) is equivalent to Eq. (13). Interestingly, it also becomes possible to have a mix of LPGA and VPGA by setting all three hyperparameters to positive values. We refer to this approach as LVPGA.

## 4 EXPERIMENTS

In this section, we evaluate the performance of LPGA and VPGA on three image datasets, MNIST (LeCun et al., 1998), CIFAR-10 (Krizhevsky & Hinton, 2009), and CelebA (Liu et al., 2015). For CelebA, we employ the discriminator and generator architecture of DCGAN (Radford et al., 2016) for the encoder and decoder of PGA. We half the number of filters (i.e., $64$ filters for the first convolutional layer) for faster experiments, while more filters are observed to improve performance. Due to smaller input sizes, we reduce the number of convolutional layers accordingly for MNIST and CIFAR-10, and add a fully-connected layer of $1024$ units for MNIST, as done in Chen et al. (2016). SGD with a momentum of $0.9$ is used to train all models. Other hyperparameters are tuned heuristically, and could be improved by a more extensive grid search. For fair comparison, $\sigma$ is tuned for both VAE and VPGA. All experiments are performed on a single GPU.

Table 1: FID scores of autoencoder-based generative models. The first block shows the results from Ghosh et al. (2019), where CV-VAE stands for constant-variance VAE, and RAE stands for regularized autoencoder. The second block shows our results of LPGA, VPGA, LVPGA, and VAE.

| Model | MNIST | CIFAR-10 | CelebA |
|---|---|---|---|
| VAE | 19.21 | 106.37 | 48.12 |
| CV-VAE | 33.79 | 94.75 | 48.87 |
| WAE | 20.42 | 117.44 | 53.67 |
| RAE-L2 | 22.22 | 80.80 | 51.13 |
| RAE-SN | 19.67 | 84.25 | 44.74 |
| VAE | $15.55 \pm 0.18$ | $115.74 \pm 0.63$ | $43.60 \pm 0.33$ |
| LPGA | $12.06 \pm 0.12$ | $55.87 \pm 0.25$ | $14.53 \pm 0.52$ |
| VPGA | $11.67 \pm 0.21$ | $\mathbf{51.51} \pm 1.16$ | $24.73 \pm 1.25$ |
| LVPGA | $\mathbf{11.45} \pm 0.25$ | $52.94 \pm 0.89$ | $\mathbf{13.80} \pm 0.20$ |

As shown in Fig. 2, the visual quality of the PGA-generated samples is significantly improved over that of VAEs. In particular, PGAs generate much sharper samples on CIFAR-10 and CelebA compared to vanilla VAEs. The results of LVPGA much resemble that of either LPGA or VPGA, depending on the hyperparameter settings. In addition, we use the Fréchet Inception Distance (FID) (Heusel et al., 2017) to evaluate the proposed methods, as well as VAE. For each model and each dataset, we take 5,000 generated samples to compute the FID score. The results (with standard errors of 3 or more runs) are summarized in Table. 1. Compared to other autoencoder-based non-adversarial approaches (Tolstikhin et al., 2018; Kolouri et al., 2019; Ghosh et al., 2019), where similar but larger architectures are used, we obtain substantially better FID scores on CIFAR-10 and CelebA. Note that the results from Ghosh et al. (2019) shown in Table. 1 are obtained using slightly different architectures and evaluation protocols. Nevertheless, their results of VAE align well with ours, suggesting a good comparability of the results. Interestingly, as a unified approach, LVPGA can indeed combine the best performances of LPGA and VPGA on different datasets. For CelebA, we further show results on 140x140 crops in Fig. 5, and latent space interpolations in Fig. 6 (see Appendix C).

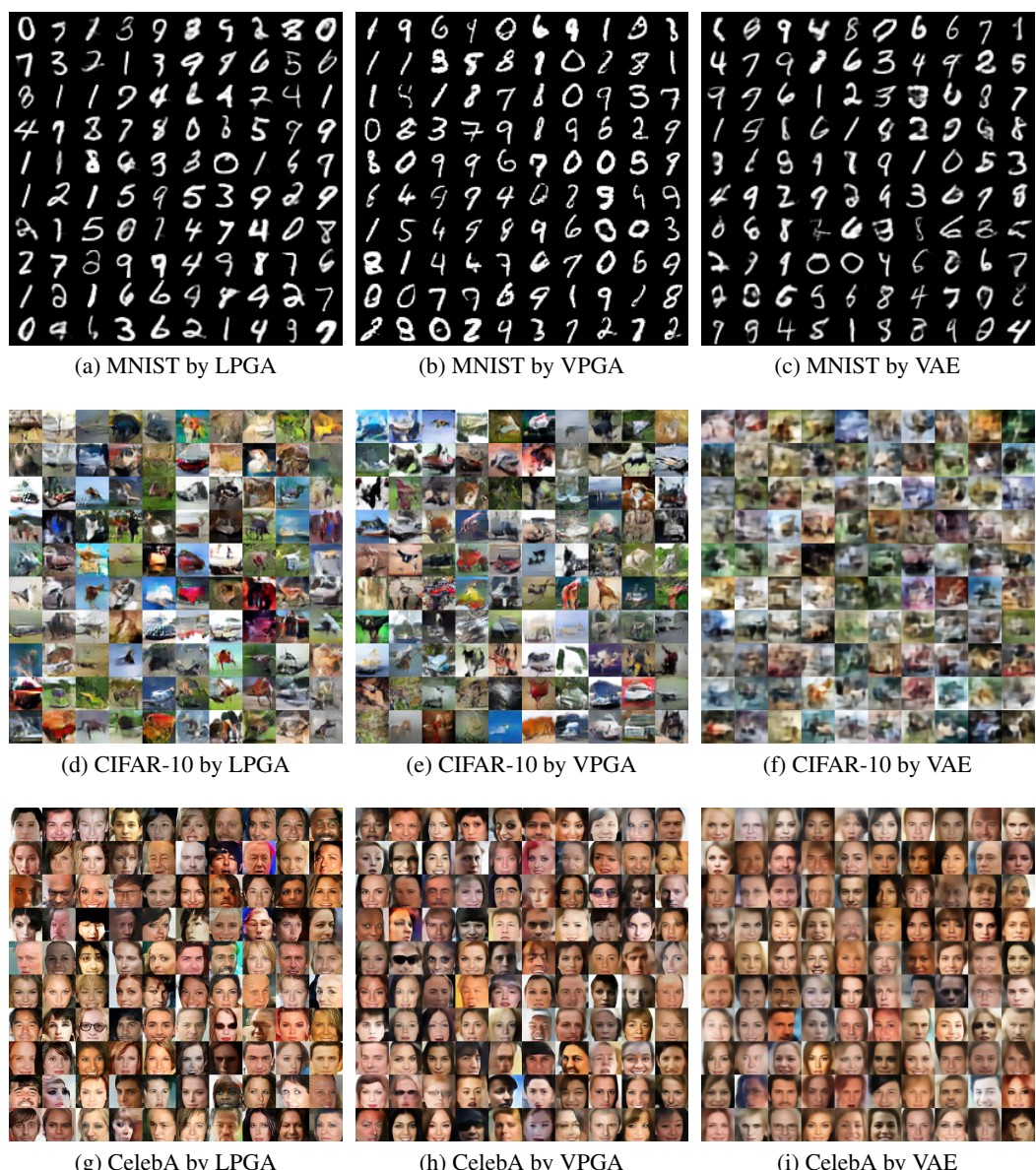

(a) MNIST by LPGA      (b) MNIST by VPGA      (c) MNIST by VAE

(d) CIFAR-10 by LPGA      (e) CIFAR-10 by VPGA      (f) CIFAR-10 by VAE

(g) CelebA by LPGA      (h) CelebA by VPGA      (i) CelebA by VAE

Figure 2: Random samples generated by LPGA, VPGA, and VAE.

The training process of PGAs is stable in general, given the non-adversarial losses. As shown in Fig. 3a, the total losses change little after the initial rapid drops. This is due to the fact that the encoder and decoder are optimized towards different objectives, as can be observed from Eqs. (4), (9), and (12). In contrast, the corresponding FIDs, shown in Fig. 3b, tend to decrease monotonically during training. However, when trained on CelebA, there is a significant performance gap between LPGA and VPGA, and the FID of the latter starts to increase after a certain point of training. We suspect this phenomenon is related to the limited expressiveness of the variational posterior, which is not an issue for LPGA.

It is worth noting that stability issues can occur when batch normalization (Ioffe & Szegedy, 2015) is introduced, since both the encoder and decoder are fed with multiple batches drawn from different distributions. At convergence, different input distributions to the decoder (e.g., $\mathcal{H}$ and $\mathcal{N}(\mathbf{0}, \mathbf{I})$) are expected to result in similar distributions of the internal representations, which, intriguingly, can be imposed to some degree by batch normalization. Therefore, it is observed that when batch normalization does not cause stability issues, it can substantially accelerate convergence and lead to

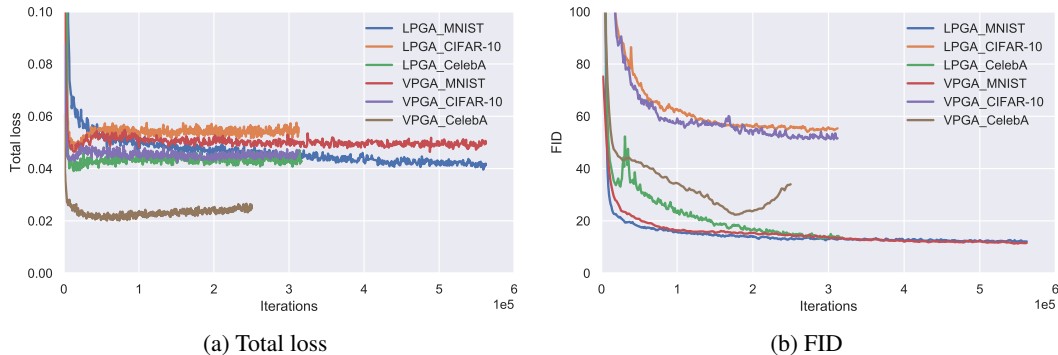

(a) Total loss
(b) FID

Figure 3: Training curves of LPGA and VPGA.

slightly better generative performance. Furthermore, we observe that LPGA tends to be more stable than VPGA in the presence of batch normalization.

Finally, we conduct an ablation study. While the loss functions of LPGA and VPGA both consist of multiple components, they are all theoretically motivated and indispensable. Specifically, the data reconstruction loss minimizes the discrepancy between the input data and its reconstruction. Since the reconstructed data distribution serves as the surrogate target distribution, removing the data reconstruction loss will result in a random target. Moreover, removing the maximum likelihood loss of LPGA or the VAE loss of VPGA will leave the perceptual generative model untrained. In both cases, no valid generative model can be obtained. Nevertheless, it is interesting to see how the latent reconstruction loss contributes to the generative performance. Therefore, we retrain the LPGAs without the latent reconstruction loss and report the results in Fig. 4. Compared to Fig. 2a, 2d, 2g, and the results in Table 1, the performance significantly degrades both visually and quantitatively, confirming the importance of the latent reconstruction loss.

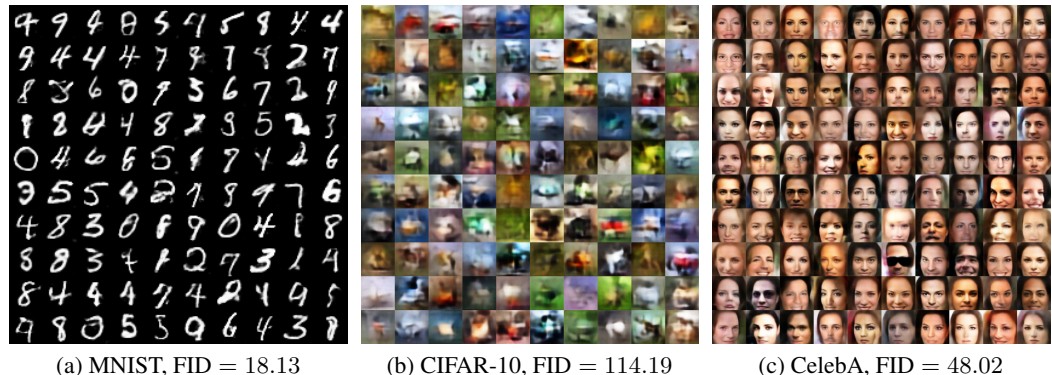

(a) MNIST, FID $= 18.13$      (b) CIFAR-10, FID $= 114.19$      (c) CelebA, FID $= 48.02$

Figure 4: Random samples generated by LPGA with $\alpha = \beta = 0$.

## 5 CONCLUSION

We proposed a framework, PGA, for training autoencoder-based generative models, with non-adversarial losses and unrestricted neural network architectures. By matching target distributions in the latent space, PGAs trained with maximum likelihood generalize the idea of reversible generative models to unrestricted neural network architectures and arbitrary latent dimensionalities. In addition, it improves the performance of VAE when combined together. Under the PGA framework, we further show that maximum likelihood and VAE can be unified into a single approach.

In principle, the PGA framework can be combined with any method that can train the perceptual generative model. While we have only considered non-adversarial approaches, an interesting future work would be to combine it with an adversarial discriminator trained on latent representations. Moreover, the compatibility issue with batch normalization deserves further investigation.

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

## A  NOTATIONS

Table 2: Notations and definitions

| | |
|---|---|
| $f_\phi/g_\theta$ | encoder/decoder of an autoencoder |
| $h$ | $h = f_\phi \circ g_\theta$ |
| $\phi/\theta$ | parameters of the encoder/decoder |
| $D/H$ | dimensionality of the data/latent space |
| $\mathcal{D}$ | distribution of data samples denoted by $\mathbf{x}$ |
| $\mathcal{H}$ | distribution of $f_\phi(\mathbf{x})$ for $\mathbf{x} \sim \mathcal{D}$ |
| $\hat{\mathcal{D}}$ | distribution of $\hat{\mathbf{x}} = g_\theta(f_\phi(\mathbf{x}))$ for $\mathbf{x} \sim \mathcal{D}$ |
| $\hat{\mathcal{H}}$ | distribution of $\hat{\mathbf{z}} = h(\mathbf{z})$ for $\mathbf{z} \sim \mathcal{H}$ |
| $L_r$ | standard reconstruction loss of the autoencoder |
| $L_{lr,\mathcal{N}}^\phi$ | latent reconstruction loss of PGA for $\mathbf{z} \sim \mathcal{N}(\mathbf{0}, \mathbf{I})$, minimized w.r.t. $\phi$ |
| $L_{lr,\mathcal{H}}^\phi$ | latent reconstruction loss of PGA for $\mathbf{z} \sim \mathcal{H}$, minimized w.r.t. $\phi$ |
| $L_{nll}^\phi$ | part of the negative log-likelihood loss of LPGA, minimized w.r.t. $\phi$ |
| $L_{nll}^\theta$ | part of the negative log-likelihood loss of LPGA, minimized w.r.t. $\theta$ |
| $L_{vr}$ | VAE reconstruction loss of VPGA |
| $L_{vkl}$ | VAE KL-divergence loss of VPGA |
| $L_{vae}$ | $L_{vae} = L_{vr} + L_{vkl}$, VAE loss of VPGA |

# B PROOFS

## B.1 THEOREM 1

*Proof sketch.* We first show that any different $\mathbf{x}$'s generated by $g_\theta$ are mapped to different $\mathbf{z}$'s by $f_\phi$. Let $\mathbf{x}_1 = g_\theta(\mathbf{z}_1)$, $\mathbf{x}_2 = g_\theta(\mathbf{z}_2)$, and $\mathbf{x}_1 \neq \mathbf{x}_2$. Since $f_\phi$ has sufficient capacity and Eq. (2) is minimized, we have $f_\phi(\mathbf{x}_1) = \mathbb{E}[\mathbf{z}_1 \mid \mathbf{x}_1]$ and $f_\phi(\mathbf{x}_2) = \mathbb{E}[\mathbf{z}_2 \mid \mathbf{x}_2]$. By assumption, $f_\phi(\mathbf{x}_1) \in Z(\mathbf{x}_1)$ and $f_\phi(\mathbf{x}_2) \in Z(\mathbf{x}_2)$. Therefore, since $Z(\mathbf{x}_1) \cap Z(\mathbf{x}_2) = \varnothing$, we have $f_\phi(\mathbf{x}_1) \neq f_\phi(\mathbf{x}_2)$.

For $\mathbf{z} \sim \mathcal{N}(\mathbf{0}, \mathbf{I})$, denote the distributions of $g_\theta(\mathbf{z})$ and $h(\mathbf{z})$, respectively, by $\widetilde{\mathcal{D}}$ and $\widetilde{\mathcal{H}}$. We then consider the case where $\widetilde{\mathcal{D}}$ and $\hat{\mathcal{D}}$ are discrete distributions. If $g_\theta(\mathbf{z}) \nsim \hat{\mathcal{D}}$, then there exists an $\mathbf{x}$ that is generated by $g_\theta$, such that $p_{\widetilde{\mathcal{H}}}(f_\phi(\mathbf{x})) = p_{\widetilde{\mathcal{D}}}(\mathbf{x}) \neq p_{\hat{\mathcal{D}}}(\mathbf{x}) = p_{\hat{\mathcal{H}}}(f_\phi(\mathbf{x}))$, contradicting that $h(\mathbf{z}) \sim \hat{\mathcal{H}}$. The result still holds when $\widetilde{\mathcal{D}}$ and $\hat{\mathcal{D}}$ approach continuous distributions, in which case $\widetilde{\mathcal{D}} = \hat{\mathcal{D}}$ almost everywhere. $\qquad\square$

## B.2 PROPOSITION 1

*Proof.* Let $\mathbf{J}(\mathbf{z}) = \partial h(\mathbf{z})/\partial \mathbf{z}$, $\mathbf{P} = [\delta_1 \quad \delta_2 \quad \cdots \quad \delta_H]$, and $\hat{\mathbf{P}} = \mathbf{J}(\mathbf{z})\mathbf{P} = [\hat{\delta}_1 \quad \hat{\delta}_2 \quad \cdots \quad \hat{\delta}_H]$, where $\Delta = \{\delta_1, \delta_2, \ldots, \delta_H\}$ is an orthogonal set of $H$-dimensional vectors. Since $\det(\hat{\mathbf{P}}) = \det(\mathbf{J}(\mathbf{z}))\det(\mathbf{P})$, we have

$$\log|\det(\mathbf{J}(\mathbf{z}))| = \log\left|\det(\hat{\mathbf{P}})\right| - \log|\det(\mathbf{P})|. \tag{17}$$

By the geometric interpretation of determinants, the volume of the parallelotope spanned by $\Delta$ is

$$\mathrm{Vol}(\Delta) = |\det(\mathbf{P})| = \prod_{i \in [H]} \|\delta_i\|_2, \tag{18}$$

where $[H] = \{1, 2, \ldots, H\}$. While $\hat{\Delta} = \{\hat{\delta}_1, \hat{\delta}_2, \ldots, \hat{\delta}_H\}$ is not necessarily an orthogonal set, an upper bound on $\mathrm{Vol}(\hat{\Delta})$ can be derived in a similar fashion. Let $\hat{\Delta}_k = \{\hat{\delta}_1, \hat{\delta}_2, \ldots, \hat{\delta}_k\}$, and $a_k$ be the included angle between $\hat{\delta}_k$ and the plane spanned by $\hat{\Delta}_{k-1}$. We have

$$\mathrm{Vol}(\hat{\Delta}_2) = \left\|\hat{\delta}_1\right\|_2 \left\|\hat{\delta}_2\right\|_2 \sin a_2, \text{ and } \mathrm{Vol}(\hat{\Delta}_k) = \mathrm{Vol}(\hat{\Delta}_{k-1}) \left\|\hat{\delta}_k\right\|_2 \sin a_k. \tag{19}$$

Given fixed $\left\|\hat{\delta}_k\right\|_2$, $\forall k \in [H]$, $\mathrm{Vol}(\hat{\Delta}_2)$ is maximized when $a_2 = \pi/2$, i.e., $\hat{\delta}_1$ and $\hat{\delta}_2$ are orthogonal; and $\mathrm{Vol}(\hat{\Delta}_k)$ is maximized when $\mathrm{Vol}(\hat{\Delta}_{k-1})$ is maximized and $a_k = \pi/2$. By induction on $k$, we can conclude that $\mathrm{Vol}(\hat{\Delta})$ is maximized when $\hat{\Delta} = \hat{\Delta}_H$ is an orthogonal set, and therefore

$$\mathrm{Vol}(\hat{\Delta}) = \left|\det(\hat{\mathbf{P}})\right| \leq \prod_{i \in [H]} \left\|\hat{\delta}_i\right\|_2. \tag{20}$$

Combining Eq. (17) with Eqs. (18) and (20), we obtain

$$\log|\det(\mathbf{J}(\mathbf{z}))| \leq \sum_{i \in [H]} \left(\log\left\|\hat{\delta}_i\right\|_2 - \log\|\delta_i\|_2\right). \tag{21}$$

We proceed by randomizing $\Delta$. Let $\Delta_k = \{\delta_1, \delta_2, \ldots, \delta_k\}$. We inductively construct an orthogonal set, $\Delta = \Delta_H$. In step 1, $\delta_1$ is sampled from $\mathcal{S}(\epsilon)$, a uniform distribution on a $(H-1)$-sphere of radius $\epsilon$, $S(\epsilon)$, centered at the origin of an $H$-dimensional space. In step $k$, $\delta_k$ is sampled from $\mathcal{S}(\epsilon; \Delta_{k-1})$, a uniform distribution on an $(H-k)$-sphere, $S(\epsilon; \Delta_{k-1})$, in the orthogonal complement of the space spanned by $\Delta_{k-1}$. Step $k$ is repeated until $H$ mutually orthogonal vectors are obtained.

Obviously, when $k = H - 1$, for all $j > k$ and $j \leq H$, $p(\delta_j \mid \Delta_k) = p(\delta_j \mid \Delta_{H-1}) = \mathcal{S}(\delta_j \mid \epsilon; \Delta_{H-1}) = \mathcal{S}(\delta_j \mid \epsilon; \Delta_k)$. When $1 \leq k < H$, assuming for all $j > k$ and $j \leq H$, $p(\delta_j \mid \Delta_k) = \mathcal{S}(\delta_j \mid \epsilon; \Delta_k)$, we get

$$p(\delta_j \mid \Delta_{k-1}) = \int_{S(\epsilon; \Delta_{k-1} \cup \{\delta_j\})} p(\delta_k \mid \Delta_{k-1}) p(\delta_j \mid \Delta_k) d\delta_k, \tag{22}$$

where $S\left(\epsilon; \Delta_{k-1} \cup \{\delta_j\}\right)$ is in the orthogonal complement of the space spanned by $\Delta_{k-1} \cup \{\delta_j\}$. Since $p\left(\delta_k \mid \Delta_{k-1}\right)$ is a constant on $S\left(\delta_k \mid \epsilon; \Delta_{k-1}\right)$, and $S\left(\epsilon; \Delta_{k-1} \cup \{\delta_j\}\right) \subset S\left(\epsilon; \Delta_{k-1}\right)$, $p\left(\delta_k \mid \Delta_{k-1}\right)$ is also a constant on $S\left(\epsilon; \Delta_{k-1} \cup \{\delta_j\}\right)$. In addition, $\delta_k \in S\left(\epsilon; \Delta_{k-1} \cup \{\delta_j\}\right)$ implies that $\delta_j \in S\left(\epsilon; \Delta_k\right)$, on which $p\left(\delta_j \mid \Delta_k\right)$ is also a constant. Then it follows from Eq. (22) that, for all $\delta_j \in S\left(\epsilon; \Delta_{k-1}\right)$, $p\left(\delta_j \mid \Delta_{k-1}\right)$ is a constant. Therefore, for all $j > k-1$ and $j \leq H$, $p\left(\delta_j \mid \Delta_{k-1}\right) = \mathcal{S}\left(\delta_j \mid \epsilon; \Delta_{k-1}\right)$. By backward induction on $k$, we conclude that the marginal probability density of $\delta_k$, for all $k \in [H]$, is $p\left(\delta_k\right) = \mathcal{S}\left(\delta_k \mid \epsilon\right)$.

Since Eq. (21) holds for any randomly (as defined above) sampled $\Delta$, we have

$$
\begin{aligned}
\log\left|\det\left(\mathbf{J}\left(\mathbf{z}\right)\right)\right| &\leq \mathbb{E}_\Delta\left[\sum_{i \in [H]}\left(\log\left\|\hat{\delta}_i\right\|_2 - \log\left\|\delta_i\right\|_2\right)\right] \\
&= H\mathbb{E}_{\delta \sim \mathcal{S}(\epsilon)}\left[\log\left\|\hat{\delta}\right\|_2 - \log\left\|\delta\right\|_2\right].
\end{aligned}
\tag{23}
$$

If $h$ is a multiple of the identity function around $\mathbf{z}$, then $\mathbf{J}\left(\mathbf{z}\right) = C\mathbf{I}$, where $C \in \mathbb{R}$ is a constant. In this case, $\hat{\Delta}$ becomes an orthogonal set as $\Delta$, and therefore the inequalities in Eqs. (20), (21), and (23) become tight. Furthermore, it is straightforward to extend the above result to the case $\delta \sim \mathcal{N}\left(\mathbf{0}, \epsilon^2\mathbf{I}\right)$, considering that $\mathcal{N}\left(\mathbf{0}, \epsilon^2\mathbf{I}\right)$ is a mixture of $\mathcal{S}\left(\epsilon\right)$ with different $\epsilon$'s.

The Taylor expansion of $h$ around $\mathbf{z}$ gives

$$
h\left(\mathbf{z} + \delta\right) = h\left(\mathbf{z}\right) + \mathbf{J}\left(\mathbf{z}\right)\delta + \mathcal{O}\left(\delta^2\right).
\tag{24}
$$

Therefore, for $\delta \to \mathbf{0}$ or $\epsilon \to 0$, we have $\hat{\delta} = \mathbf{J}\left(\mathbf{z}\right)\delta = h\left(\mathbf{z} + \delta\right) - h\left(\mathbf{z}\right)$. The result follows. $\qquad\square$

## C  More Results on CelebA

In Fig. 5, we compare the generated samples and FID scores of LPGA and VAE on 140x140 crops. In this experiment, we use the full DCGAN architecture (i.e., 128 filters for the first convolutional layer) for both LPGA and VAE. Other hyperparameter settings remain the same as for 108x108 crops. In Fig. 6, we show latent space interpolations of CelebA samples.

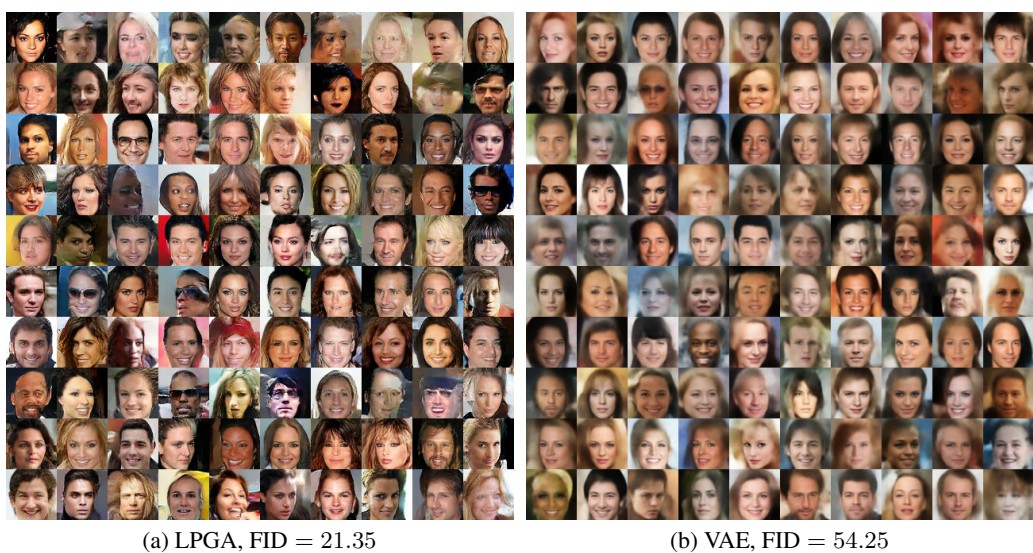

(a) LPGA, FID $= 21.35$        (b) VAE, FID $= 54.25$

Figure 5: Random CelebA (140x140 crops) samples generated by LPGA and VAE.

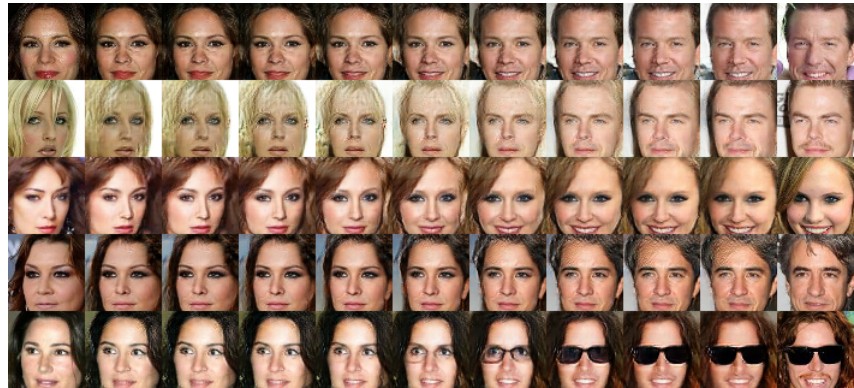

(a) Interpolations generated by LPGA.

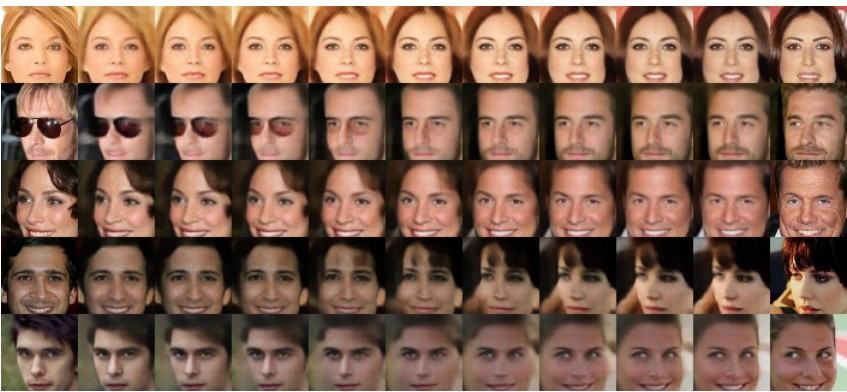

(b) Interpolations generated by VPGA.

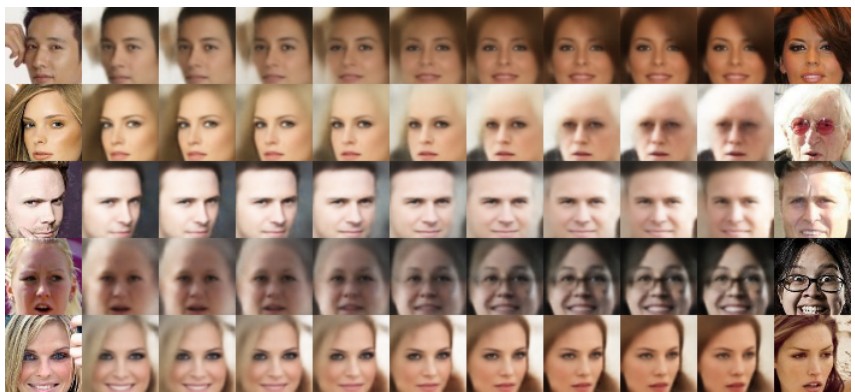

(c) Interpolations generated by VAE.

Figure 6: Latent space interpolations on CelebA.

