# OpenReview forum: "Perceptual Generative Autoencoders"
_ICLR.cc/2020/Conference — Reject_

### Official Review · AnonReviewer2 · 2019-10-23
**Official Blind Review #2**

**Rating:** 3

**Review:**

The paper considers generative models and proposes to change the VAE approach in the following way. While VAEs assume a generative model (prior on the latent space + stochastic generator) and aim at learning its parameters so as to maximize the likelihood of the (training) data distribution, the authors propose to derive the learning objective from a different view. In my understanding, they consider the two composed mappings generator(encoder) and encoder(generator) and require the first one to have the data distribution as a fixpoint and the second one to have the latent prior as a fixpoint. Starting from this idea they derive an objective function for the case that the mappings are deterministic and then further enrich the objective by either likelihood based terms or VAE based terms. The new approach is analysed experimentally on MNIST, CIFAR and CelebA datasets. The authors report quantitative improvements in terms of Frechet inception distance (FID) as well as sharper samples (when compared to standard VAEs).


I find the main idea of the paper highly interesting and compelling. Nevertheless, I would not recommend to publish the paper in its present state . The technical part is in my view very hard to comprehend. This is partially due to the disadvantageous notation chosen by the authors. Furthermore, the derivation of the individual terms in the objective is hard to understand and the arguments given in the text are not convincing:
- The two terms in the objective related to the fixpoint of the encoder(generator) mapping seem to enforce a fixpoint that is a mixture of the latent Gaussian prior and the encoder image of the data distribution. It remains unclear to me, why this is a good choice.
- The derivation of the additional likelihood based terms and VAE based terms is in my view hard to understand.
It should be possible (and I believe, is possible) to derive a simpler objective ab initio, starting from the main idea of the authors.

Side note: A close visual inspection of the presented generated samples seems to confirm the known doubts about the appropriateness of the FID measure for evaluation of generative models.

**Experience Assessment:**

I have read many papers in this area.

**Review Assessment: Checking Correctness Of Derivations And Theory:**

I assessed the sensibility of the derivations and theory.

**Review Assessment: Checking Correctness Of Experiments:**

I assessed the sensibility of the experiments.

**Review Assessment: Thoroughness In Paper Reading:**

I read the paper at least twice and used my best judgement in assessing the paper.

---

> ### Author Response · Authors · 2019-11-11
> **Response to Review #2**
>
> Thank you for appreciating the idea and for the constructive comment.
>
> - "The two terms in the objective related to the fixpoint of the encoder..."
>
> Theorem 1 requires that different $\mathbf{x}$'s generated by the decoder (from $\mathcal{N}\left(\mathbf{0},\mathbf{I}\right)$ and $\mathcal{H}$) are mapped to different $\mathbf{z}$'s by the encoder. We achieve this by training the encoder to map the output of the decoder back to its corresponding input. In theory, minimizing Eq. 2 on $\mathcal{N}\left(\mathbf{0},\mathbf{I}\right)$ would suffice, since $\mathcal{N}\left(\mathbf{0},\mathbf{I}\right)$ is supported on the whole latent space. However, there can be $\mathbf{z}$'s with low probabilities in $\mathcal{N}\left(\mathbf{0},\mathbf{I}\right)$, but with high probabilities in $\mathcal{H}$ that are not well covered by the former. Therefore, while it is not a necessity, minimizing Eq. 3 on $\mathcal{H}$ can sometimes improve the performance.
>
> - "The derivation of the additional likelihood based terms and VAE based terms is in my view hard to understand."
>
> In Sec. 3.2 and 3.3, we consider the encoder and decoder together as a generative model, $h=f\circ g$. Consequently, similar to their original forms, both the maximum likelihood loss and the VAE loss can be used to train $h$ to map the prior distribution, $\mathcal{N}\left(\mathbf{0},\mathbf{I}\right)$, to the latent-space target distribution, $\hat{\mathcal{H}}$.
>
> - "It should be possible (and I believe, is possible) to derive a simpler objective ab initio, starting from the main idea of the authors."
>
> Thank you for the suggestion. However, we note that all the loss terms are theoretically motivated, and each serves a different purpose. We explain how they work closely together in another comment.
>
> We will improve the clarity and readability of Sec. 3 in the next update.

---

### Official Review · AnonReviewer3 · 2019-10-29
**Official Blind Review #3**

**Rating:** 3

**Review:**

This paper presents a set of losses to train auto-encoders using a stochastic latent code (PGA). The authors relate it to VAE and propose a variational variant of their framework (VPGA), but the initial intuition is distinct from VAEs.

Results are presented on MNIST, CIFAR10 and CelebA and show both qualitative and quantitative improvements over VAEs.

The intuitions behind this framework seem sound and the authors added theoretical justifications when possible.

This paper seem to present a good idea that should be published, but is currently not clear enough. The writing of Section 3, especially section 3.1, need more work. The wording needs to be improved, maybe some notations are not needed, such as $\hat{z} = h(z)$. There should be a clear description of what each loss is aiming to achieve. Currently, this is not clear. For instance, there are several mentions that h must map N(0, 1) to $\hat{H}$, but loss (2) makes it map N(0, 1) to N(0, 1) (and I don't see other losses that would make it happen). It is likely I didn't understand it fully and a clearer section would help.
Another example of possible improvement would be calling the encoder and decoder with consistent names (for instance, Enc and Dec) could help, too. Currently, they are called encoder and decoder or generator in the text, f and g in the math, and theta and phi on figure 1.

I am ready to change my rating after the rebuttal if the authors address clarity of section 3.

**Experience Assessment:**

I have published in this field for several years.

**Review Assessment: Checking Correctness Of Derivations And Theory:**

I assessed the sensibility of the derivations and theory.

**Review Assessment: Checking Correctness Of Experiments:**

I assessed the sensibility of the experiments.

**Review Assessment: Thoroughness In Paper Reading:**

I read the paper at least twice and used my best judgement in assessing the paper.

---

> ### Author Response · Authors · 2019-11-11
> **Response to Review #3**
>
> Thank you for appreciating the idea and for the constructive comment.
>
> Mapping $\mathcal{N}\left(\mathbf{0},\mathbf{I}\right)$ to $\hat{\mathcal{H}}$ is achieved by either the maximum likelihood loss or the VAE loss described in Sec. 3.2 and 3.3. It can be seen by considering the encoder and decoder together as a generative model, $h=f\circ g$. Consequently, similar to their original forms, both the maximum likelihood loss and the VAE loss can be used to train $h$ to map the prior distribution, $\mathcal{N}\left(\mathbf{0},\mathbf{I}\right)$, to the latent-space target distribution, $\hat{\mathcal{H}}$. In another comment, we summarize what each loss term achieves to give an intuition of how they work together.
>
> Thank you for the suggestion on improving the wording and notation of Sec. 3. We will revise Sec. 3 accordingly.

---

### Official Review · AnonReviewer4 · 2019-11-08
**Official Blind Review #4**

**Rating:** 3

**Review:**

The main contribution of the paper is in the novel training procedure in which the target distribution and the generated distributions are mapped to a latent space where their divergence is minimised.  Training procedures for the same based on MLE and VAE are presented as well.

The idea is quite interesting but there are certain problems in clarity regarding the loss terms and the precise need for them. It appears that the work is proposing a novel framework for training VAEs where instead of comparing the target and generated distribution in the data space (the standard $L_r$), the loss is minimised by minimizing the divergence between these distributions after mapping them onto a latent space. To achieve this objective the author’s design two loss terms in addition to the standard reconstruction error. Authors provided two approaches for their training method based on MLE and VAE. Their framework is then experimentally evaluated on the standard dataset and demonstrate superior results; better-generated images and lower FID score) and provide ablation results.


The explanation of the Equation 2 is confusing, where the Figure-1a and the equation seem to agree but the subsequent description says that $h(.)$ should map ${N}(0, I)$ to $\hat{H}$ but Equation-2 will bring $h(z)$ closer to ${N}(0, I)$. Is there something missing here? The notation and the derivation of the MLE approach in Section 3.3 are not clear. This section would need rewriting with clearer and compact notation for better readability. The main idea is quite clear but how this training procedure achieves that is not coming out clearly in this version. Thus I feel the paper, though has good content, is not publishable in the current format.


**Experience Assessment:**

I have read many papers in this area.

**Review Assessment: Checking Correctness Of Derivations And Theory:**

I assessed the sensibility of the derivations and theory.

**Review Assessment: Checking Correctness Of Experiments:**

I assessed the sensibility of the experiments.

**Review Assessment: Thoroughness In Paper Reading:**

I made a quick assessment of this paper.

---

> ### Author Response · Authors · 2019-11-11
> **Response to Review #4**
>
> Thank you for appreciating the idea and for the constructive comment.
>
> As stated in the comment, Eq. 2 trains the encoder to bring $h\left(\mathbf{z}\right)$ closer to $\mathcal{N}\left(\mathbf{0},\mathbf{I}\right)$ for $\mathbf{z}\sim\mathcal{N}\left(\mathbf{0},\mathbf{I}\right)$. The purpose of Eqs. 2 and 3 is to ensure that matching the target distribution in the latent space implies the matching in the data space (Theorem 1). Mapping $\mathcal{N}\left(\mathbf{0},\mathbf{I}\right)$ to $\hat{\mathcal{H}}$, the latent-space target distribution, is achieved by either the maximum likelihood loss or the VAE loss described in Sec. 3.2 and 3.3. We will clarify this point in the manuscript. We also explain how the loss terms work together in another comment.
>
> We will improve the clarity and readability of Sec. 3 in the next update.

---

### Author Response · Authors · 2019-11-11
**How the loss terms work together**

1. Data reconstruction loss (Eq. 1):
For Theorem 1 to hold, we need to use the reconstructed data distribution, instead of the original data distribution, as the target distribution. Therefore, minimizing the data reconstruction loss ensures that the target distribution is close to the data distribution.

2. Latent reconstruction loss (Eqs. 2 and 3):
The encoder is reused to map data-space distributions to the latent space. As shown by Theorem 1, minimizing the latent reconstruction loss (w.r.t. the parameters of the encoder) ensures that if the generated distribution and the target distribution (in the data space) can be mapped to the same distribution (in the latent space) by the encoder, then the generated distribution and the target distribution are the same.

3. Maximum likelihood loss (Eqs. 6 and 7) or VAE loss (Eq. 12):
The decoder, $g$, and encoder, $f$, together can be considered as a generative model, $f\circ g$. We then train $f\circ g$ to map $\mathcal{N}\left(\mathbf{0},\mathbf{I}\right)$ to the latent-space target distribution by minimizing either the maximum likelihood loss or the VAE loss.

The first loss allows to use the reconstructed data distribution as the target distribution. The second loss transforms the problem of matching the target distribution in the data space into matching it in the latent space. The latter problem is then solved by minimizing the third loss. Therefore, the three losses together ensure that the generated distribution is close to the data distribution.

We will include this explanation in Sec. 3.

---

### Author Response · Authors · 2019-11-15
**Revision**

Dear Reviewers,

Thank you again for the valuable comments. We have revised the manuscript to improve the clarity and readability of Sec. 3.

- Improved the writing of Sec. 3.1 to avoid confusion.

- Improved the clarity of notation and figures (e.g., $f_{\phi}$, $g_{\theta}$, Fig. 1). The encoder and decoder are now referred to in a more consistent way, respectively, by $f_{\phi}$ and $g_{\theta}$ when appropriate.

- Added a subsection (Sec. 3.4) to summarize what each loss term achieves, and explain from a high-level how they work together.

---

### Decision · Program_Chairs · 2019-12-19

**Decision:**

Reject

**Comment:**

The authors present a new training procedure for generative models where the target and generated distributions are first mapped to a latent space and the divergence between then is minimised in this latent space. The authors achieve state of the art results on two datasets.

All reviewers agreed that the idea was vert interesting and has a lot of potential. Unfortunately, in the initial version of the paper the main section (section 3) was not very clear with confusing notation and statements. I thank the authors for taking this feedback positively and significantly revising the writeup. However, even after revising the writeup some of the ideas are still not clear. In particular, during discussions between the AC and reviewers it was pointed out that the training procedure is still not convincing. It was not clear whether the heuristic combination of the deterministic PGA parts of the objective (3) with the likelihood/VAE based terms (9) and (12,13), was conceptually very sound. Unfortunately, most of the initial discussions with the authors revolved around clarity and once we crossed the "clarity" barrier there wasn't enough time to discuss the other technical details of the paper. As a result, even though the paper seems interesting, the initial lack of clarity went against the paper.

In summary, based on the reviewer comments, I recommend that the paper cannot be accepted.